# The Role of Oxidative Stress in Parkinson’s Disease

**DOI:** 10.3390/antiox9070597

**Published:** 2020-07-08

**Authors:** Kuo-Hsuan Chang, Chiung-Mei Chen

**Affiliations:** Department of Neurology, Chang Gung Memorial Hospital Linkou Medical Center and College of Medicine, Chang Gung University, Taoyuan 333, Taiwan; gophy5128@cgmh.org.tw

**Keywords:** Parkinson’s disease, oxidative stress, radical oxidative species, iron, mitochondria, neuroinflammation, antioxidant, creatine, coenzyme Q10, vitamin E, pioglitazone, melatonin, desferroxamine

## Abstract

Parkinson’s disease (PD) is caused by progressive neurodegeneration of dopaminergic (DAergic) neurons with abnormal accumulation of α-synuclein in substantia nigra (SN). Studies have suggested the potential involvement of dopamine, iron, calcium, mitochondria and neuroinflammation in contributing to overwhelmed oxidative stress and neurodegeneration in PD. Function studies on PD-causative mutations of *SNCA*, *PRKN*, *PINK1*, *DJ-1*, *LRRK2*, *FBXO7* and *ATP13A2* further indicate the role of oxidative stress in the pathogenesis of PD. Therefore, it is reasonable that molecules involved in oxidative stress, such as DJ-1, coenzyme Q10, uric acid, 8-hydroxy-2’-deoxyguanosin, homocysteine, retinoic acid/carotenes, vitamin E, glutathione peroxidase, superoxide dismutase, xanthine oxidase and products of lipid peroxidation, could be candidate biomarkers for PD. Applications of antioxidants to modulate oxidative stress could be a strategy in treating PD. Although a number of antioxidants, such as creatine, vitamin E, coenzyme Q10, pioglitazone, melatonin and desferrioxamine, have been tested in clinical trials, none of them have demonstrated conclusive evidence to ameliorate the neurodegeneration in PD patients. Difficulties in clinical studies may be caused by the long-standing progression of neurodegeneration, lack of biomarkers for premotor stage of PD and inadequate drug delivery across blood–brain barrier. Solutions for these challenges will be warranted for future studies with novel antioxidative treatment in PD patients.

## 1. Introduction

Parkinson’s disease (PD) is a common neurodegenerative disease mainly involved in the progressive loss of dopaminergic (DAergic) neurons with accumulation of α-synuclein in substantia nigra (SN) of ventral midbrain [1]. These neurons secret dopamine (DA) and play a vital role in controlling the ease and balance of movements [1]. With the fall of striatal DA level below 70–80%, the clinical presentations of PD, including bradykinesia, resting tremor, rigidity and postural instability would be developed [2]. In addition, PD also displays olfactory deficits, rapid eye movement sleep disorders, depression, constipation and impairments of cognitive functions, including by affecting cholinergic, serotonergic and noradrenergic systems [1].

The etiologies of PD remain elusive. One of the causative genetic variants for PD, mutations in *LRRK2*, accounts for a number of autosomal dominantly inherited PD [3]. Mutations in *PRKN* cause early-onset PD with autosomal recessive inheritance [3]. Other genes, such as *SNCA*, *PINK1*, *DJ1*, *ATP13A2*, *GIGYF2* and *HTRA2,* have also been identified as the causative genes for familiar and early-onset PD [3]. These genes have been implicated to be involved in the ubiquitin protein degradation pathway, oxidative stress response, cell survival, apoptosis and mitochondrial function [4].

Normal cellular functions and reactions, such as oxidative phosphorylation in mitochondria, generate free reactive oxygen species (ROS) such as hydrogen peroxide (H_2_O_2_), superoxide anion (O_2_^-^) and nitric oxide. Although ROS are essential molecules for redox signaling and cellular functions, ROS-mediated oxidative damages, such as lipid peroxidation in cell and organelle membranes, protein oxidation by cross-linking, fragmentation and carbonyl group formation, as well as DNA and RNA oxidation, occur within cells [5]. Antioxidative responses up-regulate a number of antioxidants to reduce these radicals. However, as the required endogenous antioxidants are not sufficient in PD, uncontrolled production of ROS may excessively produce non-physiological and toxic ROS levels referred to as oxidative stress.

## 2. ROS Production in the PD Brain 

The brain requires plenty of oxygen supply, and a significant amount of oxygen is converted to ROS [6]. The over-production of ROS in the brain increases oxidative stress in PD patients. Lines of evidence suggest that the DA metabolism, high levels of iron and calcium in SN, mitochondrial dysfunction and neuroinflammation contribute to the increased oxidative stress and DAergic neuronal loss in the brains of PD patients (Figure 1).

### 2.1. Dopamine

Dopamine demonstrates auto-oxidation to form dopamine quinones and free radicals, which could contribute to neurodegeneration in PD [7]. The cyclization of dopamine quinones forms aminochrome, which generates superoxide and down-regulates antioxidative nicotinamide adenine dinucleotide phosphate (NADPH) [8]. The metabolism of DA by monoamine oxidase-B (MAO-B) generates 3,4-dihydroxyphenyl-acetaldehyde, ammonia and H_2_O_2_ [9]. H_2_O_2_ in DAergic neurons reacts with Fe^2+^ to form hydroxyl radical [10,11]. Induction of MAO-B in the astrocytes leads to selective loss of DAergic neurons in SN of mice [12].

The transport and storage of DA also increase the production of ROS. The storage of DA requires the transportation through vesicular monoamine transporter 2 (VMAT2) [13]. Overexpression of VMAT2 confers protection against the toxicity generated by 1-methyl-4-phenyl-1,2,3,6-tetrahydropyridine (MPTP). DAergic neurons with inhibitions of VMAT2 are more vulnerable to oxidative stress [14]. On the other hand, dopamine transporters (DAT) are required in the reuptake of DA [15]. Inhibition of DAT also increases levels of cytosolic DA that is prone to be oxidized [16,17]. Mutations of SNCA and DJ-1 in PD patients are linked to impaired DA reuptake or storage, suggesting the role of DAT in the neuronal susceptibility to oxidative stress [18,19].

### 2.2. Iron

High levels of iron are reported in SN of PD patients [20]. Iron is an essential metal for tyrosine hydroxylase, which is required for DA synthesis [21]. Iron ions Fe^3+^ and Fe^2+^ react with superoxide and H_2_O_2_ to generate hydroxyl free radicals that could be toxic to neurons [22]. Exposure of mice to iron generates neuronal loss in SN and DA depletion in striatum, develops parkinsonian phenotypes and becomes more vulnerable to MPTP [23]. Stereotaxic infusion with iron into SN of rats leads to increased levels of iron hydroxyl radicals in striatum [24]. Administrating iron chelator to mice reduces iron levels in the brain and demonstrates neuroprotective effect against iron- or MPTP-induced neurotoxicity [25]. These results suggest that increased levels of iron in SN contribute to oxidative stress and neurodegeneration in PD [24,25].

### 2.3. Calcium

The regulation of intracellular Ca^2+^ requires ATP-dependent pumps in mitochondria, which therefore increases ROS generation [26]. In primary cultures of mouse mesencephalic DAergic neurons, oxidative stress can be enhanced by opening of L-type Ca^2+^ channels, while α-synuclein aggregates potentiate this over-production of ROS in mitochondria [27]. L-type calcium channel Ca_v_1.3 is prevalent in DAergic neurons of SN [28]. This distribution in calcium channel may explain why DAergic neurons in SN are more susceptible to oxidative stress or excitotoxicity. Isradipine, an L-type calcium channel blocker, demonstrates protective effects against α-synuclein, 6-hydroxydopamine (6-OHDA) or MPTP-induced neurotoxicity in DAergic neurons [29,30,31], supporting the association between calcium influx through L-type CA^2+^ channels and PD pathogenesis.

### 2.4. Mitochondria Dysfunction

The mitochondria are primary sites of ROS production. In complex I (nicotinamide adenine dinucleotide dehydrogenase) and complex III (cytochrome bc1), the premature electron leakage to oxygen generates O^2-^ in mitochondria [32]. Mitochondrial dysfunction leads to excessive ROS production. On the other hand, ROS are also harmful to the electron transport chain itself [33,34]. The connection between dysfunction of mitochondria and PD is firstly demonstrated by MPTP-induced parkinsonism among drug abusers [35]. In the brain, MPTP is metabolized into 1-methyl-4-phenylpyridinuim (MPP^+^), which enters into DAergic neurons by DAT. MPP^+^ selectively inhibits complex I and leads to death of DAergic neurons in SN [35,36]. In PD patients, the complex I activity in SN is also reduced [37,38,39]. Genes encoding mitochondrial proteins are also down-regulated in DAergic neurons from PD patients [40]. The mitochondrial dysfunction in PD involves mitochondrial biogenesis, fusion/fission and mitophagy [41]. Furthermore, a number of genetic mutations, such as *LRRK2*, *PRKN*, *DJ-1*, *PINK1*, *FBXO7* and *ATP13A2*, in PD patients also provide clues of mitochondrial dysfunction in PD pathogenesis (see below).

### 2.5. Neuroinflammation

Neuroinflammation, mainly contributed by microglia, is now well recognized as a prominent pathological feature and a potential source of oxidative stress in PD [42]. Microglia are highly motile phagocytes and comprise 10% to 15% of the total cells in the brain [43]. Microglia can be activated and acquire phagocytic ability by α-synuclein [44,45]. Activated microglia are significant sources of oxidative stress [43] and can produce glutamate and proinflammatory factors including tumor necrotizing factor (TNF)-α, interleukin (IL)-1β and IL-6 to promote neurodegeneration [46]. In PD patients, activated microglia in SN along with an increase of pro-inflammatory factors in the brain and cerebrospinal fluid (CSF) are detected [47,48,49]. Microglial activation is observed in cell and animal models induced by MPTP, rotenone, 6-OHDA and lipopolysaccharide (LPS) [50,51,52,53,54]. The dead neurons release oxidized lipids, proteins and DNA, all of which in turn activate microglia, forming a neurotoxic vicious cycle [55]. Since the midbrain contains more microglia, the activation of microglia could be remarkably harmful to DAergic neurons in midbrain [56]. Interestingly, genetic studies find the associations between PD and single nucleotide polymorphisms within *human leukocyte antigen (HLA)* -*DRA, DRB1, DRB5* and *DQB1* regions [57,58,59,60,61,62,63,64]. These clinical studies further support the contribution of neuroinflammation to the pathogenesis of PD.

### 2.6. Oxidative Stress in Other Neurodegenerative Disorders

Similar to PD, aggregations of disease-specific misfolded proteins are main pathological features in Alzheimer’s disease (AD), Huntington’s disease (HD) and amyotrophic lateral sclerosis (ALS). AD is characterized by the presence of senile plaques, composed of β amyloid peptides (Aβ) [65]. The aggregation of Aβ activates neuroinflammation, impairs mitochondrial function and generates ROS [66]. The long polyglutamine (polyQ) tract encoded by expanded CAG trinucleotide repeats in the exon 1 of HUNTINGTIN (*HTT*) forms intranuclear and intracytoplasmic aggregates in HD [67], while a significant amount of oxidized proteins has been found in these aggregations [68]. Various mutations of TAR DNA binding protein (*TDP-43*) are found in patients with ALS [69]. These TDP-43 mutant proteins increase oxidative stress, mitochondrial dysfunction and lipid peroxidation [70]. These findings suggest that oxidative stress participates in pathogenesis of neurodegeneration in different neurological diseases.

## 3. Gene Mutations of PD Patients Involving Oxidative Stress

### 3.1. α-Synuclein (SNCA)

Mutations of *SNCA* are discovered in familial PD patients with autosomal dominant inheritance [71]. *SNCA* encodes α-synuclein, which is abundantly accumulated in Lewy bodies in degenerated DAergic neurons of PD patients [71]. Although the function of α-synuclein remains unclear, several lines of evidence suggest its role in the generation of oxidative stress [72]. Overexpression or misfolding of α-synuclein increases ROS production [73,74] and cell susceptibility to oxidative stress [74,75,76,77]. *SNCA*-transgenic mice demonstrate increased susceptibility to MPTP and 6-OHDA [78,79,80]. DAergic neurons derived from induced pluripotent stem cells (iPSCs) from a PD patient carrying triplication of the *SNCA* demonstrate high expression levels of markers of oxidative stress and augmented susceptibility to H_2_O_2_, suggesting that the overdose of α-synuclein intrinsically changes the balance of ROS production and antioxidant activities in DAergic neurons [81]. α-Synuclein may interact with phospholipase D2 to disrupt vesicular membrane integrity and recycling, leading to reduced vesicles for DA storage [82,83]. Furthermore, the accumulations of α-synuclein within the inner mitochondrial membrane inhibit activity of complex I, resulting in mitochondrial dysfunction and increased oxidative stress [84,85,86]. As an agonist of toll-like receptor 2, oligomeric α-synuclein demonstrates the potential to activate microglia [87], leading to elevated ROS production followed by the secretion of TNF-α, IL-1β and IL-6 [45,88,89]. The neuron-to-neuron transfer and propagation of α-synuclein triggers more aggregations and cytotoxic cascades [90,91,92]. It is worth noting that increased ROS may contribute to the aggregation of α-synuclein [71], which in turn increases oxidative stress, creating another neurotoxic vicious cycle.

### 3.2. PARKIN (PRKN)

Loss-of-function mutations of *PRKN* are found in early-onset PD patients with autosomal recessive inheritance [93]. *PRKN* encodes an E3 ubiquitin ligase PARKIN, which protects neurons against α-synuclein toxicity and oxidative stress [94]. In accordance with oxidative stress, PARKIN ubiquitinates mitochondrial proteins involving mitochondrial fusion and activates mitophagy to scavenge dipolarized mitochondria [95]. Knockout of *PRKN* in *SNCA*-transgenic mice exacerbates α-synuclein-induced mitochondrial dysfunction [96]. DAergic neurons derived from iPSCs carrying *PRKN* mutations demonstrate a greater amount of ROS [97,98]. On the other hand, oxidative stress down-regulates E3 ligase activity of PARKIN [99]. In *PRKN*-overexpressing DAergic neurons, treatment with DA decreases E3 ubiquitin ligase activity of PARKIN [99]. These findings indicate that PARKIN indirectly regulates oxidative stress and mitochondrial quality by mitophagy, while oxidative stress diminishes its activity.

### 3.3. PTEN-Induced Putative Kinase 1 (PINK1)

Mutations in *PINK1*, encoding a mitochondria-targeted kinase, are discovered in familial PD patients with autosomal recessive inheritance [95]. PINK1 is a key regulator of mitochondrial function by stabilization of cristae and control of mitophagy [95]. PINK1 deficiency leads to shortening, bulging and fragmentation of mitochondria [100,101,102,103,104] and loss of complex I activity [105,106,107]. Knockdown of *PINK1* in SH-SY5Y cells diminishes mitochondrial membrane potential, reduces mitochondrial DNA synthesis and ATP production [106]. *PINK1* knockout mice demonstrate an increased number of larger mitochondria with impaired function in striatum and increased susceptibility to H_2_O_2_ [107]. Neurons derived from iPSCs carrying a *PINK1* mutation demonstrate increased susceptibility to MPP^+^ and H_2_O_2_ [108]. PINK1 interacts with PARKIN to regulate mitochondrial function and oxidative stress. Upon oxidative stress, PINK1 is required for the recruitment of PARKIN to mitochondria and initiate mitophagy to scavenge damaged mitochondria [109,110].

### 3.4. DJ-1

*DJ-1* encodes a multifunctional protein participating in antioxidative stress mechanisms and mitochondrial regulation [111]. Mutations in *DJ-1* are discovered in early-onset PD patients with autosomal recessive inheritance [112]. Knockdown of *DJ-1* potentiates the cell death in oxidative stress [113,114,115]. *DJ-1* knockout mice demonstrate increased susceptibility to MPTP and 6-OHDA [116,117], while overexpressing DJ-1 reduces MPTP-induced neuronal loss in SN [116,118,119]. DJ-1 regulates the activities of nuclear factor erythroid-2-related factor 2 (NRF2) [120] and VMAT2 [121]. In low oxidative stress, NRF2 forms a complex with kelch-like ECH-associated protein 1, leading to its degradation by ubiquitination [122]. In accordance with oxidative stress, DJ-1 separates NRF2 form this complex, and the translocation of NRF2 the nucleus activates antioxidative gene expressions, leading to the reduction of ROS [111]. DJ-1 also up-regulates VMAT2 expression and activity and then enhances reuptake of excess DA into synaptic vesicles [121]. Furthermore, DJ-1 could be a potent inhibitor of death-associated protein (DAXX)/apoptosis signal-regulating kinase 1 (ASK1) cell-death pathway [123]. In H_2_O_2_-treated SH-SY5Y cells and MPTP-treated mice, DJ-1 sequesters DAXX in the nucleus to prevents its interaction with ASK1, resulting in down-regulation of apoptosis induced by oxidative stress [123,124]. Pathogenic mutations of *DJ-1* lose their neuroprotective potentials against oxidative stress [125].

### 3.5. Leucine-Rich Repeat Kinase 2 (LRRK2)

Mutations in the *LRRK2* are the most common genetic causes in PD patients [93,126,127]. LRRK2 contains a resistance to audiogenic seizures (RAS) of complex proteins (ROC) G-domain and a kinase domain. A number of pathogenic mutations of *LRRK2* identified in its kinase domain increase kinase activity, causing neuronal apoptosis [126]. Inhibition of LRRK2 kinase activity protects cells against mitochondrial dysfunction [128]. Overexpression of *LRRK2* in SH-SY5Y cells generates fragmented mitochondria and increases ROS production and susceptibility to H_2_O_2_ [129,130]. DAergic neurons derived from iPSCs carrying *LRRK2* mutations are more vulnerable to MPP^+^, H_2_O_2_ and 6-OHDA [108,131]. LRRK2 facilitates the translocation of dynamin-related protein 1 to mitochondria and then induces abnormal fission of mitochondria [129,132]. Mutations in *LRRK2* kinase domain also up-regulates phosphorylation of peroxiredoxin 3 (PRDX3), resulting in reduction of peroxidase activity and over-production of ROS. Post-mortem brain analysis from patients carrying *LRRK2* G2019S mutation consistently reveals increased phosphorylation of PRDX3 [133]. 

### 3.6. FBXO7

Mutations of *FBXO7*, an adaptor in Skp-Cullin-F-box ubiquitin E3 ligase complex, are found in early-onset familial PD patients with autosomal recessive inheritance [134]. FBXO7 involves many crucial cellular functions, including mitochondria and proteasome [135]. It is also a stress-responsive protein [136]. Stress challenges with H_2_O_2_ translocate FBXO7 to mitochondria and form FBXO7 aggregates [136], which are also seen in brains of PD patients [136]. The accumulation of FBXO7 aggregates in mitochondria leads to impaired mitochondria integrity and increased generation of ROS [136]. ROS may further facilitate FBXO7 aggregation [136], resulting in a vicious cycle of FBXO7 aggregation, mitochondria impairment and ROS generation. On the other hand, FBXO7 can interact with PARKIN and promote PARKIN recruitment to mitochondria to modulate mitophagy [137]. *FBXO7* mutations induce impairment of mitophagy as well as exaggerate FBXO7 aggregation in mitochondria [137], all of which contribute to FBXO7-associated neurodegeneration in PD.

### 3.7. ATP13A2

Mutations in *ATP13A2* are discovered in sporadic PD patients and Kufor–Rakeb syndrome (KRS), a rare early-onset atypical parkinsonism with pyramidal degeneration and dementia and autosomal recessive inheritance [138]. ATP13A2 is a P-type ATPase, which plays an important role in active transportation of cations across endosomal or lysosomal membranes [139]. The levels of ATP13A2 are reduced in DAergic neurons in SN of PD patients [140]. Lines of evidence also suggest that ATP13A2 attenuates mitochondrial dysfunction. Overexpression of ATP13A2 enhances viability of SH-SY5Y cells exposed to rotenone and MPP^+^ [141,142]. Fibroblasts of KRS patients carrying *ATP13A2* mutations demonstrate enhanced mitochondrial fragmentation, reduced mitochondrial DNA integrity and decreased ATP production [143]. Olfactory neurospheres from patients carrying *ATP13A2* mutations are also vulnerable to Zn^2+^-induced mitochondrial fragmentation and depolarization, as well as demonstrate reduction of ATP synthesis [144]. Knockdown of *ATP13A2* in SH-SY5Y cells induces mitochondrial fragmentation and increases ROS production [145].

## 4. Candidate Biomarkers for Oxidative Stress in Parkinson’s Diseases

Many studies have focused on finding biomarkers that reflect oxidative stress in PD (Table 1). In addition to improve our understanding of PD pathogenesis, these molecular biomarkers may also useful in PD diagnosis and monitoring progression of neurodegeneration in PD patients.

### 4.1. DJ-1

Elevated levels of DJ-1 in CSF or plasma are reported in PD patients [146,147,151]. On the other hand, other studies demonstrate contradicting results, which show low levels of DJ-1 in CSF of PD patients [148,149,150]. Serum levels of DJ-1 in PD patients and normal controls are similar [152]. PD patients also demonstrate higher DJ-1 levels in saliva compared to normal controls [153,154]. The levels of 4-hydroxy-2-nonenal (HNE)-modified DJ-1 isoform are significantly altered in whole blood of advanced-stage PD patients [155]. In erythrocytes and urine, levels of oxidized DJ-1 are higher in PD patients compared to normal controls [156,157], suggesting that oxidized DJ-1 could be a biomarker candidate for PD. 

### 4.2. Coenzyme Q10 (CoQ10)

CoQ10 potentiates mitochondrial electron transport chain and reduces ROS production [190]. CoQ10 also scavenges free radicals to protect mitochondrial and lipid membranes against ROS [190]. In PD patients, reduction in plasma levels of total CoQ10 and increase in ratio of oxidized CoQ10/total CoQ10 are reported [158]. The ratio of reduced form/total CoQ10 in platelets is decreased in PD patients [159], suggesting the role of CoQ10 oxidation in PD pathogenesis.

### 4.3. Uric Acid

Uric acid is a potent antioxidant and may be protective against PD [191]. In cultured SN neurons from mice, uric acid prevents death of DAergic neurons caused by H_2_O_2_ or MPP^+^ [192,193]. Elevations of cerebral uric acid in 6-OHDA-lesioned rodents improve parkinsonian phenotypes [194,195]. Two clinical trials consistently suggest that higher levels of uric acid in serum are closely related to a slower progression of PD [160,161]. High levels of uric acid in CSF are also correlated to a slower rate of clinical progression [160], as well as lower changes of unified PD rating scale (UPDRS) scores [161]. PD patients with cognitive dysfunction also have lower serum levels of uric acid compared to those without cognitive dysfunction [162]. These results suggest uric acid as a protective biomarker for PD.

### 4.4. 8-Hydroxy-2’-Deoxyguanosine (8-OHdG)

8-OHdG is generated by oxidation of guanine residues and hydroxyl radicals and in DNA [196]. It is a biomarker of DNA damage due to oxidative stress [196]. 8-OHdG concentrations are selectively elevated in SN of PD patients [197]. PD patients consistently demonstrate higher 8-OHdG levels in CSF compared to normal controls [163,164]. Furthermore, the levels of 8-OHdG in CSF are positively correlated with the duration of illness and oxidized CoQ10 in total CoQ10 [164]. PD patients demonstrate increased urinary levels of 8-OHdG [165,166], which is also correlated with the motor score of UPDRS [165]. 

### 4.5. Homocysteine

Homocysteine is an intermediate product of transulfuration methylation cycle [198]. Chronic administration of homocysteine in mice causes DAergic neuronal loss in SN by inhibiting mitochondrial activity and increasing oxidative stress [199]. Homocysteine levels in plasma of PD patients are higher compared to AD and normal controls [168]. Increased homocysteine levels are also seen in CSF of patients with PD. High homocysteine levels in plasma are correlated with worse cognition in PD patients [168,200]. It is noteworthy that levodopa therapy may increase plasma homocysteine levels, while supplement with folate or vitamin B12 can reverse levodopa-induced hyperhomocysteinemia [201].

### 4.6. Retinoic Acid (RA) and Carotenoids

RA and carotenoids demonstrate antioxidative effects in cell and animal models for PD. RA attenuates 6-OHDA and MPP^+^-mediated neurotoxicity in SH-SY5Y cells [202,203]. Administration with RA agonist prevents interferon (IFN)-γ/LPS-induced DAergic neuronal loss in rat midbrain slice cultures [204]. RA levels in plasma of PD patients are reduced [169]. PD patients also demonstrate reduced levels of α- and β-carotenes and lycopene in serum, which also inversely correlated with motor part of UPDRS scores and Hoehn and Yahr stage [170].

### 4.7. Vitamin E

Vitamin E, a lipophilic antioxidant, prevents lipids from oxidative stress [205]. Plasma levels of vitamin E are reduced in PD patients [171]. However, other studies show that serum or plasma levels of vitamin E are not different between PD patients and normal controls [172,173,174,175].

### 4.8. Glutathione Peroxidase (GSH-Px), Superoxide Dismutase (SOD) and Xanthine Oxidase

GSH-Px and SOD are two important antioxidative enzymes [206]. In PD patients, current reports of GSH-Px activities demonstrate wide variability, and could be decreased [176,177] or not altered in erythrocytes [178], or increased in serum [179,180]. A few studies indicate lower SOD activities in erythrocytes of PD patients [177,181], whereas increased [182] or unchanged [179] SOD activities in plasma or serum of PD patients are also reported. The catalyzation of hypoxanthine to xanthine by xanthine oxidase generates O2^-^ and H_2_O_2_ [207]. One study shows increased blood xanthine oxidase activities in PD patients compared to normal controls [179].

### 4.9. Nuclear Factor Erythroid 2-Related Factor 2 (NRF2)

NRF2 pathway is an antioxidative signaling pathway involved in the pathogenic processes of PD. NFR2 inactivation is observed in MPTP- or 6-OHDA-treated SH-SY5Y cells or mice [98,208]. Overexpressing α-synuclein in ventral midbrain of *NRF2* knockout mice down-regulates expression levels of NRF2-downstream genes such as nicotinamide adenine dinucleotide phosphate quinone oxidoreductase-1 (*NQO1*) and heme oxygenase-1 (*HO-1*), as well as exacerbates neurodegeneration of DAergic neurons [209]. The expressions of NRF2 and NQO1 in DAergic neurons derived from iPSCs carrying a *PARKIN* mutation are also down-regulated [97]. Recently, a small clinical study demonstrates that levels of NRF2 in peripheral leukocytes are elevated in PD patients [183]. Larger studies with correlation between NRF2 pathway and clinical severity in PD are warranted to confirm the potential of NRF2 as a biomarker for PD.

### 4.10. Lipid Peroxidation Products

Lipid peroxidation disturbs membrane organization and functional impairment of proteins and DNA [210]. A number of studies demonstrate altered levels of products of lipid peroxidation, such as isoprostanes [211,212], HNE [213] and malondialdehyde (MDA) [214,215,216], in brain tissues of neurodegenerative patients. The level of HNE in CSF is elevated in PD patients [184]. High levels of MDA have been identified in plasma of PD patients [171,182,185,186], although other studies demonstrate similar serum levels of MDA levels between PD patients and normal controls [179,187]. Plasma levels of F2-isoprostanes are also elevated in PD patients [166], whereas another study demonstrates that plasma levels of F2-isoprostanes in PD patients are similar to those in normal controls [188]. PD patients also demonstrate high levels of oxidized low-density lipoproteins (LDL) compared to normal controls [189].

## 5. Potentials of Antioxidants in Treating Parkinson’s Diseases

Current treatments for PD are prescribing levodopa with aromatic 1-amino acid decarboxylase inhibitors, dopamine agonists (eg. pramipexole, ropinirole), catechol-O-methyltransferase inhibitors (e.g., entacapone), MAO-B inhibitors (e.g., selegiline, rasagiline) and anti-cholinergic medications. Although these treatments are helpful to relieve symptoms and maintain patients’ quality of life, none of them can halt or even slow neurodegenerative processes. Considering the important role of oxidative stress in the pathogenesis of PD, antioxidants could be reasonable therapeutic strategies to modify PD progression (Table 2).

### 5.1. Creatine

Creatine is crucial to maintain energy levels particularly in the brain and cardiac and skeletal muscles [217]. It also improves mitochondrial function and serves as an antioxidant directly [218]. In MPTP-treated mice, creatine administration reduces DAergic neuronal degeneration and DA depletion [219]. A number of clinical trials have been reported to assess the potentials of creatine in treating PD patients (Table 2). A randomized, placebo-controlled study shows that creatine (2–4 g/day) has no effect on UPDRS scores but may improve moods in PD patients [220]. A larger randomized, double-blind, placebo-controlled trial on 200 PD patients shows significant reduction of UPDRS scores by creatinine (10 g/day) [221]. Another small double-blind study on 20 PD patients demonstrates that creatine (5 g/day) can improve muscle strength by resistance training [222]. However, a phase III clinical trial conducted by the National Institute of Neurological Disorders and Stroke on 1741 PD patients shows negative results by taking creatine 10 g/day for at least 5 years [223]. A potential explanation for the lack of clinical benefits in creatine and other antioxidants could be that approximately 50% of DAergic neurons and 80% of striatal DA levels have already been lost when clinical features manifest [224]. The fates of the surviving neurons may have been determined and are very hard to be altered by antioxidants.

### 5.2. Vitamin E

One population-based study shows inverse association between consumption of vitamin E and incidence of PD [225]. However, this association cannot be recapitulated by other similar studies [226,227]. In an open-labeled trial, the combination of vitamin C and E may slow disease progression in early-stage PD patients [228], while the use of levodopa could be delayed by long-term treatment with vitamin E [228]. However, another study show that vitamin E cannot slow neurodegeneration, decrease cognitive decline and mortality in PD patients (Table 2) [229]. A long-standing study further suggests that vitamin E has no effects on mortality in early-stage PD patients within 8.2 years of observation [230].

### 5.3. CoQ10

Platelet levels of CoQ10 are lower in PD patients [38]. In a small study on 15 PD patients, treatment with CoQ10 (400~800 mg/day) was well-tolerated and increased plasma levels of CoQ10 in a dose-dependent manner, whereas it demonstrates no change in UPDRS scores [231]. A randomized, double-blind, placebo-controlled clinical trial on 80 early-stage PD patients shows a significant reduction in UPDRS scores dependent on doses by administration of CoQ10 (300~1200 mg/day) [232]. This potential benefit is recapitulated by other small double-blind or open-labeled trials (Table 2) [233,234]. However, large clinical studies cannot reproduce the findings of above studies. Two larger randomized, double-blind, placebo-controlled trials do not show any benefit from higher dosages of CoQ10 (1200~2400 mg/day) [235,236]. Nanoparticular CoQ10 at 300 mg/d (equivalent to CoQ10 1200 mg/day) also displays no benefit in another randomized, double-blind, placebo-controlled trial on 131 PD patients [237]. On the other hand, administration with reduced form of CoQ10 (ubiquinol-10, 300 mg/day) in a randomized, double-blind, placebo-controlled trial on 64 PD patients in Japan shows improvement of UPDRS [238].

The limited potential to penetrate BBB may reduce the concentration of CoQ10 in CNS [239,240]. Therefore a mitochondria-targeted CoQ10 analog MitoQ is designed to enhance penetrance to BBB [241]. MitoQ is modified from CoQ10 by covalent binding of lipophilic triphenylphosphonium cation and the antioxidant moiety of CoQ10 [242]. With lipophilic property and its positive charge, MitoQ is efficiently cross BBB and accumulates within mitochondria [241,242]. However, in a randomized, double-blind, placebo-controlled trial on 128 patients, MitoQ (40 or 80 mg/day) cannot reduce the progression of PD [240]. 

### 5.4. PPARγ Coactivator-1α (PGC-1α) Agonist

PGC-1α is a pivotal regulator of mitochondrial biogenesis [243]. It also up-regulates antioxidative proteins such as GSH-Px, catalase, and SOD [244]. Genome-wide analysis revealed that many PGC-1α-driven gene expressions are down-regulated in PD patients [245]. Neurons in *PGC-1α* knockout mice demonstrate increased susceptibility to MPTP [246], while DAergic neurons in *PGC-1α*-transgenic mice are resistant to MPTP-induced neurotoxicity [247]. In rodents, administration of resveratrol, a PGC-1α activator, up-regulates genes involving mitochondrial biogenesis [248,249,250] and demonstrates neuroprotection against MPTP- [247,251,252,253] and 6-OHDA-induced neurotoxicity [254,255,256]. Epigallocatechin gallate, an important component of green, up-regulates PGC-1α to improve mitochondrial function and DA neuronal survival in MPP^+^-treated PC12 cells [257]. Another PGC-1α agonist, pioglitazone, demonstrates protective potentials against DAergic neuronal loss in an MPTP-treated rat [258]. It also mitigates the reduction of DA in striatum of rotenone-treated rats [259]. However, a population-based study does not find the relationship between pioglitazone use and PD incidence among diabetic patients [260]. A randomized, double-blind, placebo-controlled trial of pioglitazone (15 or 45 mg/day) on 210 PD patients could not show a difference of UPDRS scores between treatment and placebo groups (Table 2) [261]. Of note, pioglitazone is a substrate of CYP2C8 [262], and administration of CYP2C8 inducer, such as rifampin, phenobarbital and phenytoin, could reduce its blood level and neuroprotective effects. 

### 5.5. Glutathione (GSH), GSH-Px and SOD

GSH reduces the production of ROS and provides protection from oxidative stress [206]. Decreased levels of GSH are discovered in SN of PD patients [263,264,265,266]. Therefore, restoring the level of GSH may be a strategy to prevent damages of oxidative stress in DA neurons. PD patients with intravenous infusion of GSH (1200 mg/d) demonstrates 42% decline in disability compared to vehicle-treated controls [267]. However, another study shows no significant PD improvement by intravenous administration of GSH (700 mg/d) (Table 2) [268]. A larger randomized, double-blind, placebo-controlled trial on 45 PD patients shows intranasal GSH administration (300 or 600 mg/d) displays PD improvement similar to placebo [269].

Natural antioxidants are enriched in herb medicines and may have neuroprotective effects in PD by enhancing activities of GSH-Px or SOD. For example, gypenosides, extracted from *Gynostemma pentaphyllum*, may mitigate MPTP-induced reduction of GSH and SOD activities in mouse SN [270]. Nerolidol, found in essential oils from plants, up-regulates levels of SOD and GSH in a rotenone-treated PD mouse model [271]. Quercetin, abundant in fruits and vegetables, red wine and olive oil, restores GSH levels in striatum of 6-OHDA-treated rats [272,273]. Kaempferol, a flavonol present in tea, apple, grapefruit and broccoli, up-regulates SOD and GSH-Px activities in SN of MPTP-treated mice [274]. Resveratrol [256] and hesperetin [275] up-regulate of levels of GSH as well as activities of GSH-Px and SOD in SN of 6-OHDA-treated rats. Clinical studies will be warranted to confirm the application of these natural antioxidants in treating PD.

### 5.6. NRF2 Enhancer

The critical role of the NRF2 pathway in oxidative stress points to its potential as a target for treating PD. In cell or animal models for PD, several compounds demonstrate neuroprotective effects by up-regulating the NRF2 pathway. Dimethyl fumarate, a potent NRF2 enhancer in treating multiple sclerosis, demonstrates neuroprotection against MPTP- and α-synuclein-induced neurotoxicity in mice by activating NRF2 and HO-1 [276,277]. Deprenyl, a selective MAO-B inhibitor, up-regulates NRF2 and NQO1 to protect PC12 and SH-SY5Y cells against MPP^+^-induced toxicity [278,279]. Bromocriptine, a dopamine agonist, up-regulates NRF2 and NQO1 to protect PC12 cells from oxidative damages by H_2_O_2_ [280]. Administration of synthetic NRF2 activator CDDO-MA reduces MPTP-induced DAergic degeneration, ROS production and α-synuclein accumulation in SN of mice [281]. Treatment with metallothionein-III in 6-OHDA-treated SH-SY5Y cells up-regulates NRF2 and HO-1 expression, as well as reduces ROS production and cell apoptosis [282]. Indole derivative NC001-8 also protects DAergic neurons derived from SH-SY5Y or iPSCs carrying *PRKN* mutation against MPP^+^ and H_2_O_2_-induced neurotoxicity by up-regulating NRF2 and NQO1 [98]. By up-regulating autophagy and NRF2 pathway, disaccharides, including trehalose, lactulose and melibiose, demonstrate neuroprotective effects against α-synuclein-induced neurotoxicity [283]. Several natural compounds, such as kahweol [284], luteolin [285], ginsenoside Rb1 [286], eriodictyol [287], licochalcone A [288], genipin [97] and gastrodin [289] also demonstrate neuroprotective effects by up-regulating NRF2 pathway in different PD models. Clinical trials targeting NRF2 pathway in the future may offer a possible strategy to modify neurodegeneration in PD.

### 5.7. Melatonin

Because of its amphiphilicity, melatonin passes across BBB and possesses antioxidant properties in CNS [290]. Melatonin reduces oxidative stress and prevents neuronal degeneration in the nigrostriatal pathway of MPTP-treated mice [291,292,293,294]. The neuroprotective potential of melatonin is consistently demonstrated in 6-OHDA [295], paraquat [296,297] and rotenone-treated mice [298]. However, in a small clinical trial on 18 PD patients, 4-week administration of melatonin (3 mg/day) to PD patients has no benefit on motor performance, although this treatment may improve subjective quality of sleep (Table 2) [299]. The reduced bioavailability and first-pass effect for oral administration may reduce the level of melatonin in the targeted brain region [300].

### 5.8. Iron Chelator

The accumulation of iron in SN and its close association with production of oxidative stress suggest the application of iron chelators in treating PD. In MPTP-treated mice, desferrioxamine, a common iron chelator, inhibits iron accumulation as well as normalizes the levels of hydroxyl radical and lipid peroxidation [25]. Intraventricular injection of desferrioxamine increases DA levels in striatum of 6-OHDA-treated mice [301,302]. In a placebo-controlled, double-blind trial on 40 PD patients, treatment with desferrioxamine (30 mg/kg/day) for 12 months demonstrates lower magnetic-resonance-imaging-detected iron levels in SN compared to those with treatment for 6 months [303]. Patients with 12-month treatment also display better UPDRS scores compared to those with 6-month treatment (Table 2) [303].

**Table 2 antioxidants-09-00597-t002:** Clinical trials of antioxidants in treating Parkinson’s diseases.

Antioxidant	Number of Patients (Treatment/Placebo)	Follow-up	Dosage	Route	Effect	References
Creatine						
	60 (40/20)	2 years	4 g/day	Oral	No	[220]
	134 (67/67)	1 year	10 g/day	Oral	Beneficial	[221]
	20 (10/10)	12 weeks	5 g/day	Oral	Beneficial	[222]
	1741 (874/867)	5 years	10 g/day	Oral	No	[223]
Vitamine E						
	400 (202/199)	2 years	2000 IU/day	Oral	No	[229]
Coenzyme Q10 (CoQ10)						
	28 (14/14)	4 weeks	360 mg/day	Oral	Beneficial	[234]
	80 (64/16)	16 months	300~1200 mg/day	Oral	Beneficial *	[232]
	142 (71/71)	1 year	2400 mg/day	Oral	No	[235]
	600 (397/203)	16 months	1200~2400 mg/day	Oral	No	[236]
Ubiquinol-10						
	64 (36/28)	96 weeks	300 mg/day	Oral	No	[238]
Nanoparticular CoQ10						
	131 (64/67)	3 months	300 mg/day	Oral	No	[237]
MitoQ						
	130 (89/41)	1 year	40~80 mg/day	Oral	No	[240]
Pioglitazone						
	210 (139/71)	44 weeks	15~45 mg/day	Oral	No	[261]
Glutathione						
	20 (10/10)	12 weeks	1400 mg t.i.w. for 4 weeks	Intravenous	No	[268]
	43 (28/15)	3 months	300~600 mg/d	Intranasal	No	[269]
Melatonin						
	18 (8/10)	4 weeks	3 mg/d	Oral	No	[299]
Desferrioxamine						
	40 (21 with 12-month treatment/19 with 6-month treatment)	12 months	30 mg/kg/day	Oral	Beneficial	[303]

* In the group with coenzyme Q10 1200 mg/day.

### 5.9. Mitochondria-Targeted Antioxidant

The association of mitochondrial dysfunction and production of ROS represent mitochondria as a potential target for treating PD. Cocaine- and amphetamine-regulated transcript (CART) demonstrate their potential to be localized to mitochondria and protect SH-SY5Y cells and rat cortical and hippocampal neurons against H_2_O_2_-induced oxidative stress [304]. α-Lipoic acid protects mitochondria by up-regulating GSH levels, maintaining mitochondrial membrane potentials and inhibiting ROS production in different cell models for PD [305,306,307]. Silibinin, a major constituent of milk thistle seeds, maintains function and integrity of mitochondria and inhibits apoptosis in MPP^+^-treated rats [308]. The herbal medicine chunghyuldan inhibits ROS production and apoptosis, as well as maintains mitochondrial membrane potential, in 6-OHDA-treated PC12 cells [309]. Hesperidin, a flavanone rich in citrus, maintains mitochondrial membrane potential and inhibits ROS production and apoptosis in rotenone-treated SK-N-SH cells [310]. The flavonoid baicalein maintains mitochondrial integrity and ATP production, as well as reduces apoptosis, in 6-OHDA-treated SH-SY5Y and rotenone-treated PC12 cells [311,312,313]. In catecholaminergic neurons, tyrosol in olive oil protects cells against MPP^+^-induced toxicity and apoptosis by improving ATP production and maintaining mitochondria membrane potential [314]. Curcumin from the spice turmeric reduces ROS production and keeps mitochondrial integrity and function, as well as inhibits apoptosis, in PC12 cells overexpressing α-synuclein [315]. In MES_23.5_ DAergic cells, rosmarinic acid in *Perilla frutescens* exerts neuroprotection against 6-OHDA-induced toxicity by decreasing ROS production and maintaining mitochondrial membrane potential [316]. The marine compound xyloketal B attenuates MPP^+^-induced ROS production and reduction of mitochondria membrane potential in PC12 cells and *Caenorhabditis elegans* [317]. Moreover, the carotenoid lycopene reduces ROS production and mitochondrial damages, as well as improves ATP production, in MPP^+^-treated SH-SY5Y cells and rotenone-treated rats [318,319]. More clinical studies are essential to validate the neuroprotective effects of these compounds.

## 6. Conclusion Remarks

The pathogenesis of PD remains elusive. Lines of evidence suggest accumulation of α-synuclein, DA, iron and calcium depositions, mitochondrial dysfunction and neuroinflammation generate overwhelming oxidative stress in SN of PD patients. Deciphering the functional properties of mutations in *SNCA*, *PARKIN*, *PINK1*, *LRRK2, DJ-1*, *FBXO7* and *ATP13A2* further indicates that mitochondrial dysfunction and oxidative stress play crucial roles in PD neurodegeneration. Various biomarkers of oxidative stress, such as DJ-1, CoQ10, uric acid, 8OHdG, homocysteine, retinoic acid, vitamin E and products of lipid peroxidation are aiming to improve the early diagnosis of PD, predict its progression and monitor the therapeutic efficacy. Although numerous studies in cell and animal models support the potential of antioxidants in treating PD, many of these results cannot be reproduced in clinical trials. Novel drug delivery approaches by chemical modifications [320,321], liposomes [322,323], nanoparticles [324,325] and nanoemulsions [326,327] would be helpful to efficiently deliver antioxidants to SN and other affected brain regions of PD patients. It is also important to develop useful biomarkers as surrogate endpoints for clinical trials and to identify very early PD patients, particularly in the pre-motor stage. Carrying out robust clinical trials in large populations with application of objective biomarkers and assessment of confounding factors carefully will be essential to consolidate the therapeutic potentials of antioxidants for PD.

## Figures and Tables

**Figure 1 antioxidants-09-00597-f001:**
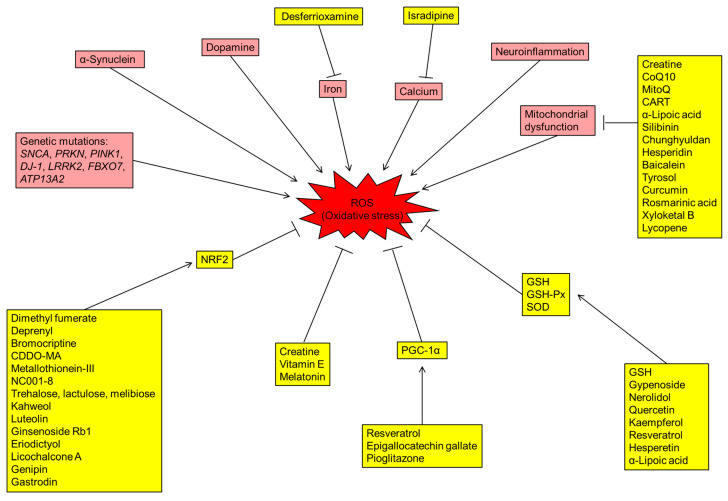
Schematic diagram representing the mechanisms of reactive oxygen species (ROS) production, antioxidative stress pathways and potential drugs with different targets for Parkinson’s disease (PD). Boxes shaded in red are indicative of genetic and environmental factors involved in the production of ROS. The drugs/proteins with potential in treating PD by reducing oxidative stress are indicated in boxes shaded in yellow. CoQ10: coenzyme Q10; CART: cocaine- and amphetamine-regulated transcript; GSH: glutathione; GSH-Px: glutathione peroxidase; NRF2: nuclear factor erythroid-2-related factor 2; PGC-1α: PPARγ coactivator-1α; ROS: radical oxidative species; SOD: superoxide dismutase.

**Table 1 antioxidants-09-00597-t001:** Potential biomarkers involving oxidative stress for Parkinson’s disease.

Candidate Marker	Origin	Change	Reference
DJ-1	CSF	↑ (PD versus NC)	[146,147]
	CSF	↓ (PD versus NC)	[148,149,150]
	Plasma	↑ (PD versus NC)	[151]
	Serum	≅ (PD versus NC)	[152]
	Saliva	↑ (PD versus NC)	[153,154]
4-HNE-modified DJ-1	Whole blood	↓ (Advanced PD versus NC)	[155]
Oxidized DJ-1	Erythrocytes	↑ (PD versus NC)	[156]
	Urine	↑ (PD versus NC)	[157]
Oxidized CoQ10/total CoQ10	Plasma	↑ (PD versus NC)	[158]
Reduced CoQ10/total CoQ10	Platelet	↓ (PD versus NC)	[159]
Uric acid	CSF	↓ (Advanced versus early stage PD)	[160]
	Serum	↓ (Advanced versus early stage PD)	[160,161]
	Serum	↓ (PD with versus without cognitive impairment)	[162]
8-OHdG	CSF	↑ (PD versus NC)	[163,164]
	Urine	↑ (PD versus NC)	[165,166]
	Plasma	↑ (PD versus NC)	[165]
Homocysteine	CSF	↑ (PD versus NC)	[167]
	Plasma	↑ (PD versus NC)	[168]
Retinoic acid	Plasma	↓ (PD versus NC)	[169]
α-carotene	Serum	↓ (PD versus NC)	[170]
β-carotene	Serum	↓ (PD versus NC)	[170]
Lycopene	Serum	↓ (PD versus NC)	[170]
Vitamin E	Plasma	↓ (PD versus NC)	[171]
	Plasma	≅ (PD versus NC)	[172,173]
	Serum	≅ (PD versus NC)	[174,175]
GSH-Px	Erythrocytes	↓ (PD versus NC)	[176,177]
	Erythrocytes	≅ (PD versus NC)	[178]
	Serum	≅ (PD versus NC)	[179,180]
SOD	Erythrocytes	↓ (PD versus NC)	[177,181]
	Plasma	↑ (PD versus NC)	[182]
	Serum	≅ (PD versus NC)	[179]
Xanthine oxidase	Serum	↑ (PD versus NC)	[179]
NRF2	Leukocytes	↑ (PD versus NC)	[183]
HNE	CSF	↑ (PD versus NC)	[184]
MDA	Plasma	↑ (PD versus NC)	[171,182,185,186]
	Serum	≅ (PD versus NC)	[179,187]
F2-isoprostanes	Plasma	↑ (PD versus NC)	[166]
	Plasma	≅ (PD versus NC)	[188]
Oxidized LDL	Plasma	↑ (PD versus NC)	[189]

↑: up-regulation; ↓: down-regulation; ≅: unchanged; CoQ10: coenzyme Q10; GSH-Px: glutathione peroxidase; HNE: 4-hydroxy-2-nonenal; LDL: low-density lipoprotein; MDA: malondialdehyde; NC: normal control; NRF2: nuclear factor erythroid 2-related factor 2; 8-OHdG: 8-hydroxy-2’-deoxyguanosine; PD: Parkinson’s disease; SOD: superoxide dismutase.

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
