# Peer review of "The Role of Oxidative Stress in Parkinson’s Disease"

_antioxidants, 2020, doi:10.3390/antiox9070597_

Round 1

Reviewer 1 Report

The authors present a review on the role of oxidative stress in Parkinson’s disease.  They first present the potential underlying mechanisms of oxidative stress in PD including auto-oxidation of dopamine, high levels of iron, changes in calcium flux, mitochondria dysfunction and neuroinflammation. Since a number of genetic mutations (alpha-synuclein, PRKN, PINK1, DJ-1, LRRK2, FBX07, ATP13A2) contribute to PD, the authors discuss how these can lead to oxidative stress. The authors then discuss different biomarkers of oxidative stress and PD that have been studied in the literature, and finish the review discussing potential use of antioxidants in treating PD, most of which are still at the preclinical stage.

There have been a number of reviews already in the literature on the mechanisms of oxidative stress in PD, but focusing on one particular mechanism.  Here, the authors present a more complete review of this timely subject, addressing the different potential modes of action involved.  In view of the numerous subjects addressed, a schematic diagram (or diagrams) would be very helpful in this type of review. For example, showing the different potential mode of actions, and where the potential therapeutics discussed would exert their anti-oxidant effects. 

The authors did publish a review on lipophilic antioxidants as applications in neurodegenerative diseases in 2018 (Clin Chimica Acta (485:79-87).  This manuscript focuses on oxidative stress in PD, but the authors present tables and data on PD very similar to that of the 2018 review.  An updated version of the review, particularly in providing schematic diagrams as mentioned above would be useful.    

In the concluding remarks (lines 511-519).  The authors bring up an interesting and important issue regarding the limits of efficacy of potential anti-oxidants, because of late diagnosis, or because of availability (BBB permeability, etc). It seems out of place in the conclusions, and it is recommended that this be moved and developed, for example, in the preceding section following presentation of the different potential anti-oxidants. 

Other comments

Line 314 “However, conflicting results in PD have also been reported [244-247].”  Please provide additional information.  How are the results conflicting? 

There are some grammatical and spelling errors, but minor enough that these can be corrected in the editing process.

Author Response

Dear Chanya:

Re antioxidants-847456 “Title: The role of oxidative stress in Parkinson’s disease”.

Sincerest thanks for your kind consideration of the publication of our article and reviewers’ comments on our manuscript. We thank the reviewers for careful and thorough reading of this manuscript and for the helpful comments and salient suggestions, which help to improve the quality of this manuscript. Please find attached the point by point reply to reviewers. Reviewer’s comments are bolded and authors’ responses are italicized. The revised sections are highlighted in yellow colour in the manuscript.

Sincerely,

Chiung-Mei Chen

Response to reviewer #1

  1. ‘A schematic diagram (or diagrams) would be very helpful in this type of review.’

Reply: We draw a schematic diagram to show the potential mechanisms of ROS production, and the potential drugs with different targets for PD.

(Line 142-148).

  1. ‘Move the limits of efficacy of potential anti-oxidants to the sections following presentation of the different potential antioxidants.’

Reply: We move these issues to preceding sections of different antioxidants.

(Line 369-372, 396-397, 419-421, 469-470)

  1. ‘ “However, conflicting results in PD have also been reported [244-247].” Please provide additional information. How are the results conflicting?’

Reply: In these studies, the serum or plasma levels of vitamin E are not different between PD patients and normal controls.

(Line 316-317)

Reviewer 2 Report

The authors try to review oxidative stress in Parkinson’s disease.

It is a well-prepared manuscript with good writing skills but still has some issues that should be addressed before accepted for publication.

Major comment

The manuscript is with 386 number of references (high in number) but only 32 references (8%) are published within five years (2015 to 2020). The latest references in the manuscript are also very few in number (2018 or 2019 or 2020; 2-3 each). The authors should give readers an update on current and emergent data in this review article. I suggest increasing the references which were published from 2015 to 2020 to >40%.

Minor comments

  1. Citing of Table 2 in-text is missing.
  2. The original names should be mentioned in the first instance with the code in brackets, for instance, the author wrote: 3.1. SNCA (α-synuclein); 3.2. PRKN (PARKIN) these should be 3.1. α-synuclein (SNCA) and 3.2. PARKIN (PRKN).

Author Response

Dear Chanya:

Re antioxidants-847456 “Title: The role of oxidative stress in Parkinson’s disease”.

Sincerest thanks for your kind consideration of the publication of our article and reviewers’ comments on our manuscript. We thank the reviewers for careful and thorough reading of this manuscript and for the helpful comments and salient suggestions, which help to improve the quality of this manuscript. Please find attached the point by point reply to reviewers. Reviewer’s comments are bolded and authors’ responses are italicized. The revised sections are highlighted in yellow colour in the manuscript.

Sincerely,

Chiung-Mei Chen

Response to reviewer #2

  1. ‘Increasing the references which were published from 2015 to 2020.’

Reply: We cited more papers (81 references) published after 2015.

  1. ‘Citing of Table 2 in-text is missing.’

Reply: In-text citations of Table 2 are added

(Line 362, 379, 389, 418, 428, 469, 480)

  1. ‘The original names should be mentioned in the first instance with the code in brackets.’

Reply: The original names in the first instance were within brackets.

(Line 151, 172, 184, 214, 274, 290, 307, 330-331)

Reviewer 3 Report

The authors wrote an extensive review on the connection between ROS and PD. The authors could consider the addition of a few lines to describe how ROS production may be worsening the phenotype of other neurodegenerative disorders as well.

Round 2

Reviewer 1 Report

Thank you to the authors for your modifications. For the Figure, it would be useful to provide a brief legend to indicate that the drugs are indicated in boxes shaded in yellow.

Author Response

Dear Sir/Madam (Reviewer 1):

Thanks for your important suggestion to improve the quality of this manuscript. We added a brief legend to describe that red boxes indicated genetic and environmental factors involved in the production of ROS, and yellow boxes showed the drugs/proteins with potential in treating PD. The revised sections are highlighted in yellow colour in the manuscript.

Sincerely,

Chiung-Mei Chen